# The Effect of Providing Staff Training and Enhanced Support to Care Homes on Care Processes, Safety Climate and Avoidable Harms: Evaluation of a Care Home Quality Improvement Programme in England

**DOI:** 10.3390/ijerph18147581

**Published:** 2021-07-16

**Authors:** Sarah Damery, Sarah Flanagan, Janet Jones, Kate Jolly

**Affiliations:** Institute of Applied Health Research, University of Birmingham, Edgbaston, West Midlands B15 2TT, UK; s.m.flanagan@bham.ac.uk (S.F.); j.e.jones@bham.ac.uk (J.J.); c.b.jolly@bham.ac.uk (K.J.)

**Keywords:** quality improvement, care home, older people, training, avoidable harms, evaluation, mixed methods

## Abstract

Older people living in care homes are at risk from avoidable harms, which may require hospital attendance or admission. This paper describes a mixed methods evaluation of a large quality improvement (QI) programme that provides skills training and facilitated support to staff in 29 care homes across two localities in the West Midlands, UK. The Safety Attitudes Questionnaire (SAQ) is used to assess changes to care home safety climate between baseline and programme end at 24 months. We use routinely collected data to assess pre- and post-programme avoidable harms and hospital attendance/admission rates. Semi-structured interviews with programme managers (n = 18), and staff (n = 49) in four case study homes are also used to assess perspectives on programme implementation. Our results show that safety climate scores increase by 1.4 points. There are significant reductions in falls (*p* = 0.0006), severe pressure ulcers (*p* = 0.014), UTIs (*p* = 0.001) and ‘any’ events (*p* = 0.0003). Emergency hospital attendances reduced, but admissions increased. Interview participants report improvements to teamwork, working practices, information sharing, knowledge and skills. Upskilling care home staff can improve working practices and attitudes towards resident safety and care quality, which may be associated with significant reductions in avoidable harms rates. Care staff turnover rates are high, which may impact the potential for longer-term sustainability of the changes observed.

## 1. Introduction

Variation in the quality and safety of care homes is growing concern for adult social care [1]. Care home residents are increasingly frail and elderly, and often have multiple physical, cognitive and sensory impairments [2]. A care home census in the UK in 2012 reported that 87% of residents have high support needs, defined as having one or more of dementia, confusion, challenging behaviour, dual incontinence, severe hearing or visual impairment or dependence on mobility [3]. In this population, adverse events can quickly escalate and lead to hospital attendance or admission [4], and high rates of avoidable harm are of particular concern across the sector [5]. The most common adverse events in care homes are accidental injuries involving residents and staff, pressure ulcers and falls [6,7].

It is commonly believed that promoting and maintaining a positive safety culture is associated with a beneficial impact on outcomes and a lower incidence of clinical and other errors that may result in patient harm [8,9]. Factors contributing to a positive safety culture include staffing levels; staff awareness of safety, training, staff willingness to improve safety and beliefs in their own ability to do so, and systems for monitoring risk and reporting adverse events [10,11,12]. A large study in 6000 care homes in the USA found that a 10% increase in staff scores on a validated patient safety culture measure was associated with lower risks of falls [4]. However, alongside the complex needs of residents, the care home sector is characterised by high workloads, high rates of staff turnover, and difficulty in recruiting and retaining competent staff [1,13]. These issues challenge efforts to introduce and embed quality improvement (QI) and positive safety practices within care home organisational culture [14]. 

Consequently, relatively few QI initiatives have been undertaken in care homes, and those that have been undertaken have often reported varying levels of success, with challenges in implementation being particularly common [6,7,15,16,17]. A recent large quality improvement programme undertaken with 90 care homes in South East England in which staff received training about QI methods and signposting to key resources found that staff knowledge and awareness of resident safety improved [18], alongside reductions in specific harms in some homes when pre- and post-programme rates were compared [19]. This paper describes the independent evaluation of a QI programme, ‘Safer Provision and Caring Excellence’ (SPACE), in which expert facilitators aimed to reduce avoidable harms and care home hospital use by providing skills training and intensive support to staff in 29 care homes across two localities in the West Midlands, UK. The evaluation used mixed methods (quantitative before-after design and in-depth qualitative case study approach), and its objectives were to describe programme implementation; assess participants’ experience, learning and changes to working practices; analyse impacts on safety climate and other outcomes, and identify the barriers to/facilitators of: (a) Effective programme implementation and (b) robust evaluation. 

## 2. Materials and Methods

This manuscript has been prepared according to SQUIRE 2.0 guidelines [20].

### 2.1. The SPACE Programme

SPACE was a complex intervention designed and implemented by Walsall and Wolverhampton Clinical Commissioning Groups (CCGs) in 29 care homes between October 2016 and September 2018. It aimed to reduce avoidable harms, accident and emergency (A&E) attendance and hospital admissions by improving safety climate and embedding QI principles into working practices at participating care homes. It was informed by the principles of Learning from Excellence (LfE) and Safety II approaches to QI [21], and comprised three interlinked components. First, managers and staff received training in QI methods from two expert facilitators. The training was open to attendance by managers and staff at all levels of seniority and in all roles, and comprised care home-based sessions focused on specific avoidable harms, and external training events. Second, facilitators supported each care home to track trends in avoidable harms using QI tools like safety crosses (a colour coded calendar on which safety incidents are recorded daily) [22], and helped in co-designing QI projects about issues most relevant to the homes themselves. Staff and managers identified the issues where they felt they needed particular support (e.g., management of pressure ulcers, information exchange during staff handover between shifts) in initial scoping meetings with project facilitators. QI projects were appraised using Plan-Do-Study-Act (PDSA) cycles, process mapping and Appreciative Inquiry (AI) [23]. Third, regular manager forums, ‘celebrating success’ events, and newsletters aimed to highlight achievements, develop a culture of continuous improvement, and create a community of best practices across participating homes. Table 1 provides a detailed description of the intervention, structured according to the Template for Intervention Description and Replication (TIDieR) guidance [24].

### 2.2. Programme Evaluation

An independent, pragmatic evaluation combining a quantitative before-after design and concurrent in-depth qualitative case study approach was undertaken to document the impacts of the SPACE programme [25]. Evaluation focus and design were guided by a programme theory [26] developed by the evaluation team following discussion with the CCGs who commissioned SPACE. The programme theory hypothesised that upskilling managers and staff would facilitate positive behaviour change and improvements to organisational safety climate, thereby improving resident safety by reducing the incidence of avoidable harms, A&E attendances and hospital admissions (Appendix A). A before-after design was chosen for the evaluation as a randomised study with contemporaneous controls could not be undertaken, and before-after designs are commonly used when evaluating QI initiatives [27]. The use of a mixed methods approach which combined quantitative analysis of prospectively collected data (care home manager and staff surveys), retrospective analysis of routinely collected data on avoidable harms, and qualitative data collection within case study care homes drew on the approach taken by Marshall et al. [18,19] in their recent study which evaluated a similar programme of QI interventions in the care home setting. In addition to the programme theory, our approach also drew on Kirkpatrick’s framework to evaluate improvement initiatives involving training and development [28]. This framework describes four levels that should be evaluated to assess whether a training-based intervention has been effective: (i) Experiences—e.g., how was the programme experienced by participants; (ii) learning—e.g., what did participants learn from the intervention; (iii) processes—e.g., how was learning used in the workplace, what changes were made to practice; (iv) outcomes—e.g., whether there were measurable improvements in safety climate and rates of avoidable harms.

### 2.3. Data Collection: Quantitative Data

Surveys were sent to all care home managers and staff regardless of role (clinical/non-clinical) or seniority at baseline, 12 and 24 months to assess care home safety climate. This was assessed using the Safety Attitudes Questionnaire (SAQ) [29], which is validated for use in the care home setting [30,31], and measures safety climate through seven questions that elicit attitudes on a five-point Likert scale. Data on respondents’ age, gender, ethnicity and role-related characteristics were also collected. Surveys were distributed internally by the home manager/administrator, and all returns were anonymous, with only the respondents’ care home identified via a barcode on each survey. Only data comparing baseline with the 24 month surveys are presented in this paper. 

Changes over time in the incidence of avoidable harms (falls, pressure ulcers, urinary tract infections (UTIs)), ambulance conveyances and hospital admissions were assessed using routine data provided monthly by each care home to their CCG. Pre-SPACE rates were compared to rates at programme end (24 months), with the pre-SPACE time period determined by data availability: Six months before baseline for avoidable harms data, and 12 months before baseline for an ambulance and hospital admissions data. Hospital data covered all admissions from the postcode in which each care home was located (c. 15 properties). Rates of staff turnover were calculated for each home in Year 1 (based on the proportion of staff employed at baseline who were no longer employed there 12 months later), and in Year 2, based on the proportion of staff employed at 12 months who were no longer employed there at 24 months. The average across these two time periods was calculated for each care home to give the staff turnover rate during SPACE. 

### 2.4. Data Collection: Qualitative Data

Four contrasting care homes—two in each geographical area, were selected as in-depth case study sites based on size and CQC ratings (Appendix B), and semi-structured interviews were undertaken with managers and staff at 12 and 24 months (49 in total) to assess programme involvement and reported changes to working practices. All staff in each case study site were eligible for interviews, and all who expressed an interest in participating were interviewed. Administrators in each home disseminated packs to their staff which included an invitation letter, Participant Information Sheet and consent form. Staff who wished to participate in an interview could contact the evaluation team by telephone, email or by mailing a postage-paid reply slip to arrange an interview. Staff focus groups were also planned in case study sites, but these were not undertaken, due to difficulties with care homes releasing multiple staff from duty simultaneously. A further 18 interviews were carried out (split equally across Years 1 and 2) with all involved CCG managers and the programme facilitators to elicit perspectives about programme implementation and changes over time. All interviews followed pre-specified topic guides and were undertaken following written consent. They were carried out face-to-face (care home staff) or by telephone (CCG managers and facilitators), audio-recorded and transcribed verbatim. The evaluation team also observed all centrally-run events and selected training sessions in individual care homes over the two years of SPACE (184 h total) to gain an overview of programme content and delivery. Training sessions were selected to ensure that each training topic (e.g., falls prevention, nutrition/hydration support, Learning from Excellence) was observed in both Walsall and Wolverhampton at least once in each care home where it was offered. Observations were documented using detailed field notes. 

### 2.5. Data Analysis

Quantitative analysis was undertaken using SPSS version 24.0 (IBM Corp., Armonk, NY, USA). Survey responses were analysed descriptively, and between-group differences were tested using chi-square for independence. Independent sample *t*-tests were used to assess differences in mean SAQ safety climate scores between baseline and programme end as group sizes differed between surveys. Missing SAQ data were imputed using a participant’s mean score for the domain. Avoidable harm, A&E and hospital data were analysed descriptively, comparing monthly incidence pre-baseline and at programme end per 100 beds. 

Qualitative data were managed using NVivo version 12.0 and analysed thematically [32]. Analysis followed Braun and Clarke’s six stage analytical process: (1) Data familiarisation; (2) initial code generation; (3) initial theme development from codes; (4) theme/code review; (5) theme definition and refinement and (6) producing an analytic narrative of the findings. Two researchers (SF, JJ) independently analysed 10% of interview transcripts to create the initial coding framework. The framework was discussed by the evaluation team, and amendments made or new codes added until all data had been analysed. Field notes from observations were also analysed thematically. Supporting quotations from interviews are referenced in the text (Q1, Q2 etc.); the corresponding quotations are provided in Appendix A.

After analysing the survey and interview data separately, data were combined where possible (e.g., in relation to safety climate) using a triangulation protocol [33] in which the main qualitative and quantitative findings were compared to identify agreement, complementary information, or contradictions. This allowed a more complete picture of the effectiveness of SPACE than would have been obtained by keeping the data entirely separate. All data in this paper relate to both geographical areas combined, unless otherwise indicated. 

## 3. Results

### 3.1. Characteristics of Participating Care Homes

A total of 29 care homes (1882 bed capacity) participated in SPACE. Each care home was either targeted by their CCG or volunteered their involvement. Twenty-six homes (16 in Wolverhampton, 10 in Walsall) consented to participate in the evaluation, all of which were privately owned and provided care for older people, and all but one were registered to provide both residential and nursing care. The number of beds per care home ranged from 12 to 84 (mean = 49). At baseline, 14/26 homes (53.8%) were classed as ‘good’ by the CQC, 11/26 (42.3%) required improvement, and 1/26 (3.8%) was ‘inadequate’.

### 3.2. Training Uptake

Over 1000 staff across all job roles and all levels of seniority received QI methods training as part of SPACE—representing an annual participation rate of around 60% of staff employed in participating care homes. Training quality was rated highly by managers and staff in the survey data, and numerous interview participants felt that the learning from training had translated into perceived improvements to practice within their care home(s) (Q1,2). 

### 3.3. Changes to Safety Climate

Qualitative data showed that SPACE was associated with perceived changes to safety climate within participating care homes. Staff reported feeling empowered to suggest and test ideas (Q3), with autonomy to implement change (Q4,5). Using data to support QI was widespread (Q6), and a clear culture of information sharing and mutual support developed both within (e.g., through staff cascading the learning from training to others) (Q7) and between care homes (Q8). Staff reported increased confidence when liaising with external agencies like tissue viability teams and continence services (Q9), and the CQC explicitly recognised that the changes made as a result of SPACE had improved quality in multiple care homes. Survey response rates were 38.6% at baseline (566/1465) and 37.1% at programme end (565/1521). There was a non-significant 1.4 point improvement in mean SAQ safety climate score between baseline and programme end (Table 2). The only statement with a statistically significant change over time-related to learning from the mistakes of others (from 80.7 to 85.1; *p* = 0.005). 

Secondary analysis, comparing mean SAQ safety climate scores by care home size (small, medium, large) and quality (as defined by the CQC overall rating at baseline) is reported in Appendix A. Aggregate mean safety climate scores improved significantly for medium sized care homes (30–49 beds) (from 83.3 to 86.5; *p* = 0.030), and for care homes with higher CQC ratings (from 84.6 to 88.0; *p* = 0.003). For individual SAQ safety climate statements, there was a significant improvement for small care homes (<30 beds) in relation to the statement ‘the culture here makes it easy to learn from the mistakes of others’ (from 79.0 to 92.4; *p* = 0.005). For several other statements, there were significant improvements in SAQ scores for care homes rated as higher quality by the CQC. These higher quality care homes improved significantly with regard to perceptions of feeling safe if participants lived at the care home (from 84.4 to 90.1; *p* = 0.0008); feeling that medical areas were handled appropriately (from 88.8 to 92.7; *p* = 0.006); that colleagues encouraged the reporting of resident safety concerns (from 87.5 to 91.2; *p* = 0.017) and that the culture at the care home made it easy to learn from others’ mistakes (from 81.4 to 87.9; *p* = 0.0004).

Sub-group analysis of the 24 month surveys showed higher safety climate scores significantly associated with job role, working full-time, being qualified beyond school level, and attending SPACE training. Scores were also significantly higher for smaller vs larger care homes, those with lower than average rates of staff turnover, and those with higher CQC quality ratings (Table 3).

### 3.4. Avoidable Harms, A&E Attendance, Hospital Admissions

There were statistically significant reductions in several avoidable harms when pre-SPACE and post-SPACE data were compared (Table 4). Monthly falls rates reduced significantly over time, as did the incidence of Grade 4 pressure ulcers, UTIs and aggregated data for ‘any’ events. There was an increase in rates of Grade 2 pressure ulcers, A&E attendances reduced over time, and rates of hospital admissions increased (all non-significant). 

### 3.5. Changes to Safety Processes

Qualitative and observational data showed widespread use within most care homes of risk monitoring tools, such as safety crosses, and there were numerous examples of generic tools being adapted internally to monitor specific quality areas within care homes (Q10). The collection and interpretation of data with support from the programme facilitators was seen as a positive way to facilitate QI and monitor change (Q11), and there was widespread engagement with QI techniques, such as PDSA cycles, LfE and AI. These techniques were reported to have improved staff attitudes towards care quality, avoiding assigning blame when an adverse event occurs, and instead considering what usually goes right when such events are avoided (Q12). The co-design of initiatives between programme facilitators and care homes was felt to empower participants to take ownership of QI at the care home level (Q13,14). There were also reports of perceived improvements to teamwork, communication and sharing of best practices, both within individual homes and across the wider network of SPACE care homes. 

### 3.6. Manager and Staff Experiences

Manager and staff interviews revealed almost universally positive attitudes towards SPACE, even when enthusiasm had been lacking at the outset (Q15). There was a feeling that programme learning had made a substantial perceived improvement to care home quality and safety (Q16,17). Participants were enthusiastic about the programme and their experience of training, and reported positive perceptions of improvements in staff autonomy, confidence and empowerment to make an effective change (Q18–20). Numerous examples of changes to practice were cited, and there was optimism about the potential legacy of SPACE (Q21,22), although there were concerns about programme sustainability after active facilitation ceased (Q23,24).

### 3.7. Challenges to Implementation

Several challenges to programme implementation were identified by programme managers and facilitators. First, there was suspicion from some managers and staff when SPACE began, perceiving that SPACE participation would entail increased bureaucracy and workloads. The facilitators reported substantial efforts in building relationships between themselves and each care home to overcome these perceptions. Second, whether SPACE training and support facilitated effective QI often depended heavily on the leadership of individual care home managers. Ten homes changed their registered manager at least once during the programme. This raised challenges for sustaining engagement and maintaining momentum in programme implementation. Third, annual rates of staff turnover averaged 31% in participating homes (range 9.6% to 78.3%): Similar to the national average across adult social care in England [34]. This may have affected how the learning from skills training could become embedded within participating care homes. However, the facilitators believed that their flexible approach to training, and the emphasis on QI project co-design, mitigated the possible negative effects of staff turnover. High priority was given to workforce development and highlighting opportunities for staff career advancement as the programme developed. 

Finally, there was variable engagement from participating homes which may have impacted how SPACE was adopted meaningfully. Facilitators reported that some care homes participated fully in all aspects of the programme and positive changes became largely self-sustaining for those homes. In contrast, a small number of care homes maintained minimal participation throughout the programme, attended few training or regional learning events, and adopted few QI tools and strategies. The remainder participated well, but needed substantial ongoing support to facilitate their involvement. The facilitators addressed these issues by modifying programme implementation over time. Whereas, Year 1 of SPACE focused on relationship building, delivering training and supporting care homes to adopt QI methodologies, Year 2 focused on consolidation, with significant programme resources directed towards engaging with care homes that had been only partially involved previously. This approach was largely successful in improving care home involvement, as shown by staff attendance at training and implementation of QI projects.

### 3.8. Challenges to Evaluation

There were some tensions between the evaluation team’s requirement to give an unbiased assessment of progress, and the desire of those delivering the programme to celebrate achievements and sustain engagement from participants. Positive findings were sometimes interpreted too positively by the programme team, and challenging results were typically viewed defensively. Moreover, it was necessary for the evaluation team to develop close relationships with the facilitators so that all elements of programme activity could be captured. This closeness created a risk that our desire to see the programme succeed could compromise our position as independent, external evaluators. These issues were reflected on regularly and discussed within the evaluation team alongside the ongoing data analysis to ensure that our independent position was maintained. The mixed methods approach and use of data from multiple sources also helped reduce the risk of bias when interpreting the collected data. 

### 3.9. Sustaining Change

Although many managers and staff were confident that SPACE had become embedded in their day-to-day working practices, some worried that the momentum would be lost when the active provision of SPACE training stopped, particularly given the high level of workforce turnover in many care homes and a potentially short ‘organisational memory’. Ensuring the sustainability of programme learning was a key focus in the latter stages of the programme, and both CCGs attempted to enhance QI capability across their respective areas and spread QI skills widely within teams that supported care homes. Support was also formally linked to planning at STP (Sustainability and Transformation Partnership) level, alongside commitments to maintain ongoing relationships between care homes and specialist teams (e.g., tissue viability) and allow care home managers and staff to access training offered by regional NHS organisations. However, given the finding that the important elements of success for SPACE were the intensive support provided by facilitators directly to care homes and the provision of bespoke training and resources specific to the needs of each care home, the removal of one-to-one support may compromise the sustainability of programme learning. 

## 4. Discussion

Identifying the ‘active ingredients’ of successful QI interventions is an important goal for current policy initiatives to enhance care within care homes [35]. This evaluation used a quantitative before-after design combined with a qualitative case study approach to describe the implementation of a large care home QI programme within the West Midlands. Overall, the SPACE programme can be considered effective—training uptake was high, managers and staff reported feeling empowered to design, implement and assess their own QI projects with support from programme facilitators, and numerous positive changes to learning, teamwork and working practices were reported. Findings from the quantitative and qualitative data analysis corroborated each other, suggesting the observed improvements could be attributable to the SPACE programme. Quality improvement programmes are always more than the sum of their individual parts, but the evaluation identified multiple success factors that contributed to programme effectiveness (Box 1). These factors may be useful for others planning similar programmes, particularly when mapped onto different levels of the system from the individual staff (micro) level, through care home structures and processes (meso level), up to the wider regulatory context in which care homes operate (macro level) (Appendix A).

Box 1Success factors for the SPACE programme.
Passionate facilitators with an in-depth understanding of issues within the care home sector, who tailored programme support accordingly.Developing ways to engage and empower a wide range of staff, not just managers or senior nurses.Intensive, ‘hands-on’ facilitation where participating care homes received multiple facilitator visits over the course of the programme, and could contact the facilitators about any issue, at any time.Focusing on the co-design of quality improvements with the care homes rather than standardised tools or approaches being implemented in a top-down manner.Having the flexibility to use language and examples relevant to care homes, and delivering tailored training that combined theory with practical application.Focusing on the use of simple rather than complex tools for facilitating QI in participating care homes.Building strong relationships with care home managers who helped to foster positive relationships within the care homes and supported staff to see that the programme was worthwhile and important.Supporting the care homes to collect and interpret their own data for quality improvement and for tracking trends over time.Providing ideas, encouragement, resources and ongoing support.Providing regular feedback on progress and encouraging care home managers and staff to develop a sense of ownership of change.Providing opportunities for care homes to share ideas, best practices and to learn from each other.Supporting care homes in their liaison with external organisations to make them feel that they were a valuable part of the wider health economy.


Baseline scores for safety climate were substantially higher than those suggested by the relevant SAQ benchmarking data [28,29], leaving little headroom for improvement following programme implementation. However, the small increases observed in the survey data suggest that SPACE contributed to maintaining high levels of positive safety climate perception amongst staff in participating homes. This is particularly important given that high staff turnover made it likely that most staff completing the Year 2 surveys were not the same individuals that responded at baseline. This suggests that positive attitudes towards safety were becoming embedded within participating care homes, although care homes that were rated as higher quality at baseline appeared to have more receptive cultures in which improvements to safety climate could be made, as shown by the statistically significant increases in SAQ scores from baseline to 24 months seen in secondary sub-group analyses of safety climate.

Analysis of routinely collected data showed a significant reduction over time in rates of falls, severe (Grade 4) pressure ulcers, UTIs and the incidence of ‘any’ events. Rates of A&E attendance reduced slightly, and hospital admission rates increased slightly, but neither were statistically significant. The likelihood that relatively short-term, training-based QI programmes can demonstrate substantial changes to ‘hard’ outcomes like hospital use may be disputed, yet QI programme managers typically have high expectations for rapid progress in addressing such outcomes [36,37]. The relationship between care home quality improvements and hospital use may be complex: QI skills training may make care home staff more risk-averse, making them more likely to send a resident to the hospital if there were doubts over their health or observed rate of deterioration. Conversely, staff may become less likely to ask for a resident to attend the hospital if they can recognise the early signs of issues like pressure ulcers and manage them effectively internally. Moreover, a relatively high proportion of A&E attendances, hospital admissions and harms like falls are unavoidable given the nature of the care home resident population [38]. Unfortunately, without a detailed case note review, we could not judge whether or not a given event could be considered avoidable. 

Quality improvement projects may fail through misunderstanding the behavioural and cultural changes needed to support improvement [36]. Care home staff often have low literacy levels, and English may be a second language, which presents challenges for designing training and motivating staff to attend [39]. The pragmatic approach taken by the programme facilitators in tailoring training to the audiences seems to have been effective in overcoming these challenges. SPACE also focused on facilitating positive changes to working culture by applying approaches that emphasised the role of leadership, capacity building and sharing of best practices. The co-design element of programme activities between facilitators and care home staff was reported as particularly important, and has been noted by others as a key means of embedding QI within organisations where active engagement from managers and staff is required for practice changes to be implemented [38,40,41]. Indeed, the influence of care home leadership on QI is often cited in the patient safety literature [42], and other similar programmes have found managerial leadership and care home capacity to engage with QI to be important influences on uptake and involvement [19].

The context within which care homes operate is fundamentally important to how improvement initiatives are implemented and evaluated. Relationships between local authorities, CCGs and care homes are complex, and competing priorities may mean that improvement initiatives are not always given the necessary time and focus. There has been much local, regional and national interest in SPACE, and resources/toolkits have been made available to other areas planning QI interventions in care homes. However, there may be challenges in reproducing successful interventions in new contexts, particularly where interventions are multifaceted and tailored to their context of origin [43]. Thus, it is unlikely that simply sharing the resources from SPACE with those in other areas will lead to successful implementation elsewhere [44]. The issue of context is also influenced by past efforts to improve quality in the care home sector. It is likely that some of the effectiveness of SPACE was related to the legacy of concerted efforts to improve quality in care homes for a number of years prior to SPACE being developed. As a result, many participating care homes had longstanding relationships with their local CCG, ensuring that they were largely receptive to engagement with SPACE. Other areas without this history may find it more difficult to replicate the success of SPACE. Similarly, it is well known that changes take time to become embedded, and we cannot know whether the changes observed as part of SPACE are entirely attributable to the programme itself, or whether they reflect the maturation of changes that have come about following ongoing attempts to improve quality and safety in care homes within the region. 

### 4.1. Strengths and Limitations

This evaluation was robustly conducted, and methodological triangulation ensured that findings relating to safety climate and programme impact were corroborated across data sources. There was a high level of involvement from care home staff in both the programme and our evaluation. Nevertheless, before and after designs may overestimate intervention effects [45]. We had anticipated using an algorithm developed by Nottingham University to extract data on hospital use directly from the relevant hospital Trusts and from matched control care homes in a separate Trust in the West Midlands. This would have indicated whether observed trends in hospital use were secular or could be attributed to the programme. However, obtaining research permissions was challenging and could not be completed within the evaluation timeframe. Consequently, hospital admission rates may have been overestimated: Identifying admissions from care homes using postcodes is imprecise, and some admissions from neighbouring properties may have been erroneously included. The routine data analysed to assess trends in harms over time varied in quality and completeness, with few participating care homes reporting data for every month pre-programme or during SPACE. No objective measure of manager or staff behaviour change could be included; thus, our understanding of such change is based on self-reported evidence from the interviews and/or surveys. Nevertheless, the data obtained from the quantitative data (surveys, avoidable harms); from the observations of training sessions and events regionally and in individual care homes, and the rich data obtained through participant interviews across the case study care homes suggest a strong likelihood that the observed improvements in safety climate and reductions in avoidable harms are attributable to the QI programme. It would be unlikely that both the quantitative and qualitative data would have corroborated each other if the programme had not been effective. 

### 4.2. Implications for Policy and Practice

Well-designed QI programmes that are implemented flexibly; where participants co-design QI projects with expert facilitators, and where care homes are intensively supported to share ideas and use data to drive and appraise changes can lead to widespread improvements to working practices and staff perceptions of safety. Our evaluation adds to a growing body of evidence that multi-component interventions with a strong basis in techniques like LfE and AI can facilitate meaningful participation by care home staff despite financial, political and workforce pressures within social care. In conducting a mixed methods evaluation, our work gives a clear idea of enablers and barriers to implementation, as well as potential success factors of relevance to other, similar QI programmes [46]. In reporting detailed SPACE programme aims and activities using a structured template, we attempt to enhance the clarity and replicability of SPACE activities (and evaluative work) by others [47,48]. 

### 4.3. Implications for Research

The lack of a nationally-agreed minimum dataset for UK care homes is a recognised limitation for social care research. The COVID-19 pandemic strongly highlighted the need for more reliable identification of the care home population within national routine datasets as well, and there are studies underway to address this [49,50]. The availability of such datasets would increase the scope and potential robustness of evaluative studies that could be carried out to assess the potential effectiveness of future QI programmes, perhaps using step-wedge or other randomised designs in which the effects of an intervention could be compared with contemporaneous matched controls. With regard to SPACE itself, a key focus for future work will be a post-evaluation assessment of avoidable harm and hospital use rates for the 2–3 years following the phase of active SPACE implementation. Evaluations rarely capture the longer-term outcomes of QI initiatives, and this planned follow-up work will allow us to assess the longevity and potential sustainability of the changes observed and reported here.

## 5. Conclusions

The provision of bespoke and flexible QI skills training and intensive facilitator support to participating care homes as part of the SPACE programme positively impacted managers and staff, working practices, and on care homes’ collaborations with each other and across the wider health economy. The flexible approach taken to programme implementation was a key strength, and there were trends towards meaningful reductions in avoidable harms. This suggests that change had become embedded within participating care homes despite high rates of staff turnover and the inherent challenges associated with the complex health and care needs of the care home resident population. Further post-programme evaluation may determine how longer-term change was sustained.

## Figures and Tables

**Table 1 ijerph-18-07581-t001:** Detailed description of the SPACE quality improvement programme.

Domain	Description of Design and/or Activities Undertaken
Name:	Safer Provision and Caring Excellence (SPACE) Programme.
Why:	To upskill staff and promote a culture of continuous quality improvement (QI) that could reduce avoidable harms in participating care homes.
Where:	Walsall and Wolverhampton, West Midlands, UK.
Who provided:	Intervention delivered by two full-time facilitators (one in each area) experienced in QI. Appreciative Inquiry (AI) workshops to support positive safety culture were delivered by an external provider (https://www.appreciatingpeople.co.uk, accessed on 24 February 2021). Programme was funded by the West Midlands Academic Health Sciences Network Patient Safety Collaborative.
Recipient(s):	Twenty-nine care homes: Eleven in Walsall; eighteen in Wolverhampton. Intervention administered to managers, senior/junior nursing and care staff, staff in domestic, administrative and maintenance roles, activity coordinators.
How:	Face-to-face meetings and training.
When and how much:	Twenty-four months (December 2016 to December 2018). Eight half/full-day shared learning events (4 in Walsall and 4 in Wolverhampton). Monthly training in participating care homes on specific topics for managers and staff. One to one coaching/support from facilitators throughout the programme.
What: (materials and procedures)	Collaborative shared learning events: Networking: Exhibition stalls promoting harm free care resources (e.g., tissue viability); stalls run by regional/national training providers (e.g., My Home Life, Skills for Care). Skills development via group training and breakout sessions on harm-specific and general QI topics (e.g., PDSA cycles). Invited speakers gave overviews of national/regional challenges faced by the sector and facilitators presented on SPACE progress. Care homes presented their QI projects and their ‘improvement journey’. Education and training: Training delivered by facilitators or specialist teams through small groups, or larger training workshops attended by staff from several care homes. Training delivered on: Leadership and culture: Engaging stakeholders, managing change, safety culture, human factors training. Measurement for improvement: Model for Improvement Driver Diagrams conceptualised QI and design projects based on SMART aims, choice and measurement of outcomes, PDSA cycles to assess improvement effectiveness. Communication/handover: Optimising staff shift changes to support safety culture, e.g., safety boards to highlight risks visually and minimise the risk of errors and harms. *Workforce development*: Training attendees asked to identify their learning from each session and describe how they would cascade that learning to colleagues to facilitate changes in care home practice. Support from facilitators: Facilitators visited each home to give ad hoc support and one-to-one QI coaching. This included reviewing PDSA data on specific QI projects, co-developing action plans, signposting to resources/training opportunities, helping to collect and interpret data, risk and harm monitoring, and providing data run charts to capture trends over time. Care homes also supported providing evidence to the Care Quality Commission (CQC). Recognition/sharing of best practices: Bi-monthly newsletters, highlighting achievements, sharing learning, advertising training, signposting to useful resources. Care home managers and staff also provided content (e.g., photos and articles describing events held at their home). Annual awards ceremony to ‘celebrate success’ and recognise the innovative practice. Bi-monthly forums for managers, led by facilitators. Designed to build relationships, develop shared purpose, provide peer support and share best practices. Programme sustainability: Resource toolkit and best practices guidelines developed. Facilitator role in Wolverhampton integrated into the Quality Nurse Advisor (QNA) role and QNA officers trained in QI. In Walsall, QI nurses undertake joint quality visits with the local authority. Evaluation: Independent evaluation of SPACE development, implementation and effectiveness undertaken by a regional university.
Tailoring:	Shared learning events used Appreciative Inquiry principles, focusing on what works well, and human factors to understand errors. Programme elements aligned with local and national priorities and best practices, e.g., CQC domains of care and hospital avoidance. Flexible design and delivery of care home-based training integrating lessons learned from incidents, mapping exercises to encourage staff groups (maintenance, domestic etc.) to identify their own contribution towards particular aims. Training events designed to elicit a ‘commitment to act’ from attendees and cascade learning to others to improve practice.
Programme changes:	Flexibility: One-to-one coaching support to managers, small group training in the care home, larger workshops with staff from multiple care homes, larger collaborative events to disseminate and share learning. Responsiveness: Training modified throughout to respond to feedback and focused on topics identified by homes as areas of interest. Adaptation: Events were linked with specialist clinical training from clinical partners, e.g., falls, tissue viability, dementia. Underpinning theories: Inclusion in Year 2 of human factors principles; recognising and managing deteriorating residents. Co-design: Emphasis on co-design of QI projects between facilitators and managers/staff. Workforce development: Opportunities for continuing professional development and career advancement were highlighted to staff.

**Table 2 ijerph-18-07581-t002:** Comparison of mean SAQ safety climate scores between baseline and 24 months.

Statement	Baseline ^1^	24 Months	Comparison ^3^
I would feel safe if I lived at this care home	83.3	86.1	*p* = 0.063
Medical areas are handled appropriately	88.6	90.8	*p* = 0.079
I know who to ask about resident safety	93.0	93.1	*p* = 0.930
I receive appropriate feedback about my performance	81.9	81.0	*p* = 0.590
It is difficult to discuss errors in this care home ^2^	69.4	68.8	*p* = 0.785
My colleagues encourage me to report any resident safety concerns	86.7	88.5	*p* = 0.209
The culture here makes it easy to learn from the mistakes of others	80.7	85.1	*p* = 0.005
Aggregate mean score for domain	83.4	84.8	*p* = 0.179

^1^ Higher scores = greater agreement with statement; ^2^ Statement is reverse scored; ^3^ Comparisons used independent *t*-tests.

**Table 3 ijerph-18-07581-t003:** Sub-group analysis of mean SAQ safety climate scores at 24 months.

Variable	Grouping	Mean Score ^1^	Comparison ^2^
Age	18 to 24	82.1	
	25 to 34	84.8	
	35 to 44	86.8	*p* = 0.271
	45 to 54	82.7	
	55 to 59	85.5	
	60+	86.3	
Gender	Male	85.6	*p* = 0.786
	Female	85.0	
Ethnic group ^3^	White	85.1	*p* = 0.516
	BAME	84.3	
First language	English	84.9	*p* = 0.453
	Non-English	83.6	
Job role	Manager	86.7	
	Nurse	90.1	
	Care assistant	82.6	*p* < 0.0001
	Domestic/Kitchen	83.6	
	Maintenance	87.0	
	Administration	93.9	
	Activity co-ordinator	93.5	
Working hours	Full-time	86.8	*p* = 0.043
	Part-time	83.6	
Shift pattern	Day staff/mixed shifts	85.5	*p* = 0.370
	Evenings/nights only	81.5	
Qualifications	School level education	84.1	*p* = 0.006
	Education beyond school level	88.4	
SPACE training attendance	No training	80.6	*p* < 0.0001
	Centrally-organised OR care home	85.9	
	Centrally-organised AND care home	89.1	
Care home size	Small (<30 beds)	90.2	*p* < 0.0001
	Medium (30–49 beds)	86.5	
	Large (50+ beds)	80.0	
Staff turnover ^4^	Lower than average	86.8	*p* = 0.017
	Higher than average	83.2	
CQC rating	Outstanding/Good	88.0	*p* < 0.0001
	Requires improvement/Inadequate	79.2	

^1^ Higher score = more positive perception of safety; ^2^ Comparison used independent *t*-tests; ^3^ BAME = Black, Asian and Minority Ethnic; ^4^ Annual staff turnover was calculated for each care home and care homes categorised as lower or higher than the aggregated mean across all homes participating in SPACE.

**Table 4 ijerph-18-07581-t004:** Pre- and post-SPACE avoidable harms, A&E attendance, hospital admissions.

	PRE-SPACE ^1^		SPACE ^2^		Comparison ^3^
	Events/beds ^4^	Rate/100 beds/month	Events/beds	Rate/100 beds/month	
Falls	442/4205	10.5	1713/20342	8.4	*p* = 0.0006
Pressure ulcer (Gr2)	87/8731	1.0	140/11611	1.2	*p* = 0.14
Pressure ulcer (Gr3)	46/8731	0.5	52/11611	0.5	*p* = 0.41
Pressure ulcer (Gr4)	27/8731	0.3	18/11611	0.2	*p* = 0.014
Urinary Tract Infections	25/4205	0.6	58/19754	0.3	*p* = 0.001
ANY event	547/4205	13.0	2213/22042	11.0	*p* = 0.0003
A&E attendance	881/10134	8.7	2184/25495	8.6	*p* = 0.699
Hospital admission	287/9125	3.1	722/20049	3.6	*p* = 0.052

^1^ Avoidable harms data cover 6 months pre-SPACE, A&E attendance and hospital admission data cover the 12 months before SPACE; ^2^ ‘SPACE’ data cover the 24 months of programme implementation; ^3^ Comparisons calculated using chi-square; ^4^ Bed numbers vary due to differences in data availability by month (see Appendix B).

## Data Availability

There are ethical issues limiting the availability of data generated and analysed in this study. Qualitative data cannot be publicly shared even if de-identified, due to concerns over participant confidentiality and privacy, and due to the terms of participant consent, as noted by the ethics committee that approved the study. Excerpts of interview transcripts relevant to the study are available from the research governance office of the University of Birmingham (researchgovernance@contacts.bham.ac.uk). Data on avoidable harms, A&E attendances and hospital admissions from participating care homes are not available as these data are owned by Walsall and Wolverhampton Clinical Commissioning Groups and not the evaluation team. The survey dataset is also unavailable, due to ethical concerns that individual care homes could be identified from their data.

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
