# Peer review of "The Effect of Providing Staff Training and Enhanced Support to Care Homes on Care Processes, Safety Climate and Avoidable Harms: Evaluation of a Care Home Quality Improvement Programme in England"

_ijerph, 2021, doi:10.3390/ijerph18147581_

Round 1
Reviewer 1 Report
1.- Please present literature or previous works discussion. The authors present a Statistical methodology but there is no theoretical or previous works foundation.
There must be similar papers that study similar issues with this or other methods. Please discuss them and relate them briefly to your paper. This, to give support of what are you doing and why are you doing it.
2.- In the discussion of the results, the authors suggest that the actual tests do not give strong proofs to tell that the SPACE program was indeed a real cause for improvements. Please discuss or suggest what ar your conclusions and usefulness of SPACE valid to strengthen your review.
3.- Related to the previous suggestion, please present guidelines for further research and extensions of your work.
Author Response
We would like to thank the reviewers for their comments on our manuscript, and we are pleased that it was viewed positively. We appreciate the opportunity to revise the manuscript to provide further clarity about our approach. Please see our point-by-point response to the reviewer comments below.
- Please present literature or previous works discussion. The authors present a Statistical methodology but there is no theoretical or previous works foundation. There must be similar papers that study similar issues with this or other methods. Please discuss them and relate them briefly to your paper. This, to give support of what are you doing and why are you doing it.
Response: Thank you for this comment, and apologies that we were not clear enough about our chosen evaluation method. Extra text has been added to the ‘programme evaluation’ section (page 3, lines 103-118) explaining our approach. Briefly, we chose to do a before-after, mixed methods evaluation which is a standard approach for evaluating QI initiatives where a randomised design with contemporaneous controls is not possible. The mix of qualitative and quantitative methods follows best practice guidance for evaluation design as it allows multiple aspects of a programme of work to be evaluated through different ‘lenses’ and tracked over time. Full detail on the evaluation design and its justification is given in our published protocol which is referenced in the manuscript (Reference 25). Our choice of evaluation methods also follow a recent similar care home QI project by Marshall et al. that we reference in our background, where the intervention being evaluated comprised many of the same features as the SPACE programme (References 18,19), and the theoretical framework for evaluating training-based interventions, developed by Kirkpatrick and Kirkpatrick (2006) (Reference 28).
- In the discussion of the results, the authors suggest that the actual tests do not give strong proofs to tell that the SPACE program was indeed a real cause for improvements. Please discuss or suggest what ar your conclusions and usefulness of SPACE valid to strengthen your review.
Response: Thank you for this comment. We do not feel that our manuscript lacks a strong sense of what conclusions we can draw from our study and regarding the effectiveness of SPACE - our apologies if this seems the case. Our entire discussion section describes the findings and our review of the robustness of findings derived from each of our data collection methods, both separately and when considered together through triangulation. We are obliged to acknowledge the limitation that without a control group that didn’t receive the intervention, it is impossible to be 100% sure that the observed changes were due to the intervention. However, this is a common limitation of before-after study designs and being unable to be sure without doubt that observed changes are directly attributable to an intervention is not unusual. We do not feel that this makes our findings (or our interpretation of them) any less meaningful or valid. Indeed, using a mixed methods approach shows a high degree of likelihood that the changes we saw were due to the intervention – there is a consistency of findings with regard to improvements in safety climate, substantial reductions in some avoidable harms, and in the self-reported changes to practice and understanding obtained from the case study work. The case studies in particular demonstrate richness of data, and it is clear that the participants thought that the positive changes that happened were definitely because of the intervention. It is unlikely that both the quantitative and qualitative data would have shown complementary findings if there was not an underlying positive association between the intervention and the observed outcomes.
We have added a summary statement within the strengths and limitations section that strengthens our overall view about the robustness of our findings and the extent to which we feel the observed positive changes were attributable to the QI intervention programme itself (page 14, lines 497-504).
- Related to the previous suggestion, please present guidelines for further research and extensions of your work.
Response: We have added a new paragraph under the heading ‘implications for research’ describing future research plans (page 15, lines 532-545). Part of this paragraph includes text about the lack of national-agreed minimum datasets which has been moved from the limitations section; the remainder of the paragraph focuses on potential future work within the care home sector and in relation to SPACE itself.
Reviewer 2 Report
First the CCG is to be commended for implementing this project. The paper is well written and gives a balanced view of the challenges to working in this sector.
My only suggestion is to present the results using several criteria.
Perhaps summarise the before and after results by banding the care homes by initial CQC assessment, by size (bed numbers) and perhaps also by most improved/least improved.
This additional assessment can be included in a modified form of Tables 2 to 4. As always the use of charts can be an effective way of communicating key results.
Other than these suggestions, well done on an excellent paper.
Author Response
We would like to thank the reviewers for their comments on our manuscript, and we are pleased that it was viewed positively. We appreciate the opportunity to revise the manuscript to provide further clarity about our approach. Please see our point-by-point response to the reviewer comments below.
1. First the CCG is to be commended for implementing this project. The paper is well written and gives a balanced view of the challenges to working in this sector. My only suggestion is to present the results using several criteria. Perhaps summarise the before and after results by banding the care homes by initial CQC assessment, by size (bed numbers) and perhaps also by most improved/least improved. This additional assessment can be included in a modified form of Tables 2 to 4. As always the use of charts can be an effective way of communicating key results.
Response: Thank you for these suggestions. We were able to do some of what the reviewer asks; unfortunately we could not do this for the avoidable harms data because of the inflexible format in which the data were provided to us. First, we have undertaken additional analysis to break the data presented in Table 2 (comparison of mean SAQ safety climate scores by individual safety climate statement and in aggregate) down into sub-groups by a) care home size – small, medium and large, and b) by CQC rating of care home quality at baseline – ‘outstanding/good’ vs. ‘requires improvement/inadequate’.
As the above constitute secondary analyses, we have chosen to display the additional data in a standalone table as part of our supplementary files (Table S2), rather than modifying the existing table 2 to accommodate the data on care home size and quality, because the latter would turn the current 8 row table into a rather cumbersome 48 row table. We are also mindful of the need to maintain conciseness in the manuscript so as not to increase its overall length too much. We would of course be happy to amalgamate Table 2 and Table S2 and place the combined table within the main manuscript if this was deemed appropriate by the editors. Some additional text has been added to section 3.3 to highlight the statistically significant results that Table S2 shows.
Table 3 (sub-group analysis of mean SAQ safety climate scores at 24 months) already included data broken down by care home size and quality, so no further changes have been made to this table.
Although it would have been interesting to analyse the data in Table 4 (avoidable harms data) by care home size and quality, unfortunately we were unable to do this because of the format of the data. Unlike the SAQ data, which we collected ourselves, the data on harms and hospital use were provided to us by a third party (the CCG), in an anonymised and aggregated format which we could not break down further in order to assign specific care homes to specific sub-groups. Undertaking sub-group analysis on the harms data was not an aim of the evaluation (due to the likely small monthly event rate), so only aggregated data were made available to the evaluation team.
2. Other than these suggestions, well done on an excellent paper.
Response: Many thanks for this positive feedback, we are pleased that you felt that the manuscript is of high quality.
Round 2
Reviewer 1 Report
The authors satisfy the review suggestions properly.